# Effect of Ionic Conductors on the Suppression of PTC and Carrier Emission of Semiconductive Composites

**Yingchao Cui [1], Hongxia Yin [1], Zhaoliang Xing [2], Xiangjin Guo [1], Shiyi Zhao [1], Yanhui Wei [1], Guochang Li [1], Meng Xin [1], Chuncheng Hao [1,2,*] and Qingquan Lei [1]**

[1] Institute of Advanced Electrical Materials, Qingdao University of Science and Technology, Qingdao 266042, China; cc18753214515@163.com (Y.C.); 13455018936@163.com (H.Y.); Guoaiwei525@163.com (X.G.); zsy19941103@126.com (S.Z.); weiyhui@126.com (Y.W.); Lgc@qust.edu.cn (G.L.); xinmeng_7591@126.com (M.X.); leiqingquan@qust.edu.cn (Q.L.)

[2] State Key Laboratory of Advanced Power Transmission Technology (Global Energy Interconnection Research Institute Co., Ltd.), Beijing 102209, China; xingzhaoliang007@163.com

[*] Correspondence: clx@qust.edu.cn; Tel.: +86-187-0532-1299

**Abstract:** The positive temperature coefficient (PTC) effect of the semiconductive layers of high-voltage direct current (HVDC) cables is a key factor limiting its usage when the temperature exceeds 70 °C. The conductivity of the ionic conductor increases with the increase in temperature. Based on the characteristics of the ionic conductor, the PTC effect of the composite can be weakened by doping the ionic conductor into the semiconductive materials. Thus, in this paper, the PCT effects of electrical resistivity in perovskite $La_{0.6}Sr_{0.4}CoO_3$ (LSC) particle-dispersed semiconductive composites are discussed based on experimental results from scanning electron microscopy (SEM), transmission electron microscopy (TEM) and a semiconductive resistance test device. Semiconductive composites with different LSC contents of 0.5 wt%, 1 wt%, 3 wt%, and 5 wt% were prepared by hot pressing crosslinking. The results show that the PTC effect is weakened due to the addition of LSC. At the same time, the injection of space charge in the insulating sample is characterized by the pulsed electroacoustic method (PEA) and the thermally stimulated current method (TSC), and the results show that when the content of LSC is 1 wt%, the injection of space charge in the insulating layer can be significantly reduced.

**Keywords:** $La_{0.6}Sr_{0.4}CoO_3$; semiconductive layer; PTC effect; space charge; HVDC transmission

## 1. Introduction

High-voltage direct current (HVDC) transmission plays a significant role in the power system [1–5]. In particular, HVDC cable transmission is feasible over long distances and large capacities due to the absence of reactive power and low transmission losses [6,7]. Typical medium and high-voltage power polyethylene (PE)cable cross-sectional constructions include: (1) conductors, (2) conductor shield, (3) insulation, (4) insulation shield, (5) metal shield, and (6) enclosure material [8]. In the construction of high-voltage power cables, the semiconductive layer can suppress the injection of carriers from the metal electrode into the insulating layer and can effectively prevent local electric field distortion between the conductor and the insulating layer.

However, the electrical resistance of the semiconductive layer can suddenly increase to 90 °C, which causes the cable to heat up and leads the interface to partially melt. This phenomenon is called the positive temperature coefficient (PTC) effect [9]. The PTC effect of semiconductive composites is usually weakened by increasing the content of carbon black (CB) or by using high-structure carbon black [10,11]. However, the amount of CB added to the semiconductive shielding layer affects its

processing and mechanical properties. The conductivity of the ionic conductor increases with increasing temperature. In this work, the influence of the (CB—$La_{0.6}Sr_{0.4}CoO_3$ (LSC)) co-filled on the electrical properties of semiconductive composites was studied, in which LSC was used as a second filler to suppress the PTC effect. The perovskite oxide $LaCoO_3$ has been widely used because of its high ionic and electrical conductivity. The ideal perovskite structure is shown in Figure 1 [12–14]. When La in $LaCoO_3$ is partly replaced by the Sr, the lattice spacing becomes larger and the oxygen vacancies in the crystal increase [15]. Oxygen vacancies and lattice defects of LSC can provide more conductive channels for electrons, which facilitates electron migration when Sr-doped $LaCoO_3$ is added to a semiconductive composite material.

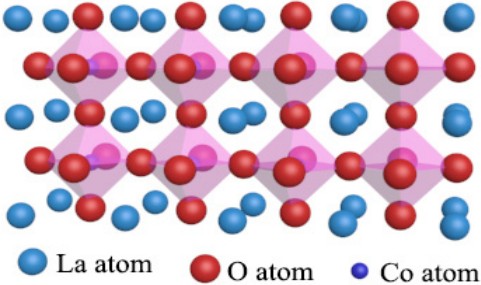

**Figure 1.** Structure of $LaCoO_3$.

On the other hand, space charge accumulation of the insulating layer is another key factor affecting the stable operation of the cable. Insulating layers tend to accumulate space charge, which causes distortion of the electric field [16,17]. Eventually, the insulating layer is easily aging or mangled [18–20]. Until recently, most studies have been limited to the insulation nano-doped polyethylene, which has been attracting more and more attention. It has been reported that polyethylene-doped inorganic nanoparticles such as MgO, ZnO and $SiO_2$ can significantly suppress the accumulation of space charge in the insulation [21–25]. Some researchers add $SrFe_{16}O_{19}$ to the semiconductive layer to reduce the injection of space charge in the insulating layer by using the Lorentz force of magnetic particles on the charge [26]. However, in this study, we suppressed the injection of space charge by using the LSC modified semiconductive layer. The injection of charges into the insulating layer is reduced, through the Coulomb effect between LSC particles in semiconductive materials and injected charges. This work provides a new idea for the development of semiconductive materials.

## 2. Materials and Methods

### 2.1. Materials

#### 2.1.1. Preparation of LSC

The LSC was prepared by using the sol-gel method [27,28], mixing a stoichiometric amount of lanthanum nitrate hexahydrate $La(NO_3)_3\cdot6H_2O$ strontium nitrate $Sr(NO_3)_2$ with cobalt nitrate hexahydrate $Co(NO_3)_2\cdot6H_2O$ in deionized water under constant stirring to get a clear solution. Citric acid (CA) was then added into the solution (CA and total metal ion in a 7:5 molar ratio), in which as a ligand to form a complex compound with the metal ion. Then, the pH value of the solution was adjusted to 9–10 by dropwise addition of aqueous ammonium hydroxide. The solution was slowly evaporated in a water bath at 70 °C for 10 h and the gel obtained was at the temperature of 150 °C overnight. Finally, the obtained powder was calcined at 900 °C for 6 h to obtain LSC nanoparticles.

#### 2.1.2. Ball Milling of LSC

The prepared LSC powder was ball milled in a planetary ball mill, where 10 g of LSC powder and zirconia balls (mass ratio of zirconia balls to LSC powder of 20:1) were added to a ball mill jar, and

then 200 mL of ethanol were added. The speed during ball milling was 300 rpm. The samples of ball milling of 20 and 40 h, respectively, were obtained for observation by scanning electron microscopy. Finally, the samples were dried at 50 °C to obtain the LSC nanoparticles after ball milling [29].

### 2.1.3. Preparation of the Nanocomposite

The matrix polymer was prepared by mixing 25% carbon black (CB), 45% low-density polyethylene (LDPE), and 30% ethylene-vinyl acetate copolymer (EVA) with an open mill at °C. Then, the LSC was mixed with the above matrix polymer in different mass percentages, as shown in Table 1. At last, the above materials were shaped by hot pressing by a vulcanizer.

**Table 1.** Sample notation and composition.

| Sample | 1# | 2# | 3# | 4# | 5# |
|---|---|---|---|---|---|
| CB/LDPE/EVA matrix (wt%) | 100 | 99.5 | 99 | 97 | 95 |
| LSC (wt%) | 0 | 0.5 | 1 | 3 | 5 |

### 2.2. Characterization

#### 2.2.1. X-ray Diffraction (XRD)

The crystal structural analyses of LSC were determined by XRD measurements (Rigaku, D/max-2500/PC) from 20° to 90°.

#### 2.2.2. Scanning Electron Microscopy (SEM)

The morphology of the nanoparticles was observed with a field emission SEM (FEI·Nova·Nano·SEM450) at a 5 kV accelerating voltage. Dispersion of the nanoparticles in the semiconductive composites was observed using SEM (JSM-6700F, JEOL, Tokyo, Japan). The nanocomposites were broken in liquid nitrogen and then fractured cross-sections were sprayed with gold to avoid the charge accumulation effect during observation.

#### 2.2.3. Transmission Electron Microscopy (TEM)

In order to characterize the dispersion of the LSC in the matrix polymer, 50–100 nm thick ultra-thin sections were cut using a ultramicrotome and observed using a TEM (FEI·Tecnai·G2F30).

#### 2.2.4. Resistivity Test

In the actual operation of the cable, the working temperature is greatly affected by the load. The resistance of the semiconductive layer will increase with the increase in the temperature, showing obvious PTC effect, which will lead to the increase in the interface thermal effect between the semiconducting layer and the insulating layer, and affect the service life of the cable. In this work, the resistivity of the semiconductive layer was measured by the DB-4 wire and the cable semiconductive rubber-resistance tester using the (DC) current-voltage method test principle. The samples, with length 110 mm, width 50 mm, and thickness of 1 mm, were obtained by hot pressing crosslinked. The sample is placed in a drying oven with programmable temperature control, and the resistivity of the sample is recorded at different temperatures. When the instrument is used to measure, the sample does not need surface treatment, and the operation is simple. The resistivity of the sample can be obtained directly without formula derivation and calculation, thus avoiding the error in the calculation process.

#### 2.2.5. Pulsed Electroacoustic Measurement (PEA)

The distribution of space charge was tested by PEA. The LDPE insulating sample used for PEA testing had an average thickness of 300 μm and the semiconductive layer had a thickness of 500 μm.

The experiment was carried out for 30 min at room temperature under a negative DC electric field of 10 and 40 kV/mm. The PEA test chart is shown in Figure 2.

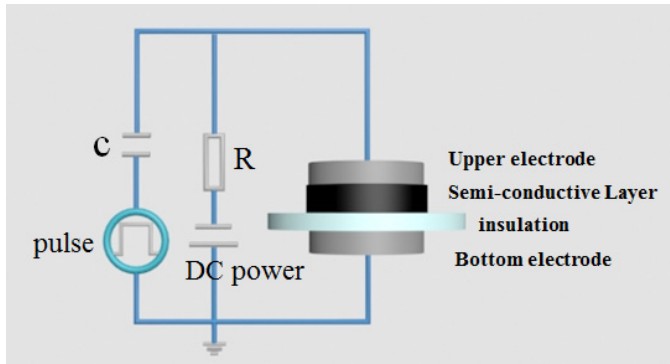

**Figure 2.** Illustration of the pulsed electroacoustic (PEA) measurement.

### 2.2.6. Thermally Stimulated Current (TSC)

The TSC method includes the thermal stimulation polarization current method (TSPC) and the thermal stimulation depolarization current method (TSDC). The TSDC method is more common in the measurement and characterization of traps in polymer insulation. Thus, the TSC method generally refers to the TSDC method. The TSDC method was used in this experiment. The thickness of the insulating sample and the semiconductive layer used for TSC was 300 and 500 μm, respectively. A negative DC field strength of 10, 30, and 40 kV/mm was applied to both ends of the LDPE for 30 min at room temperature when the semiconductive composites with different LSC contents were used as the semiconductive layer, and then, the sample was rapidly cooled. Next, the temperature was raised from 293 K at a heating rate of 5 K/min to 363 K to measure the value of the thermal stimulation current during the heating process. The schematic diagram of the TSDC test is shown in Figure 3.

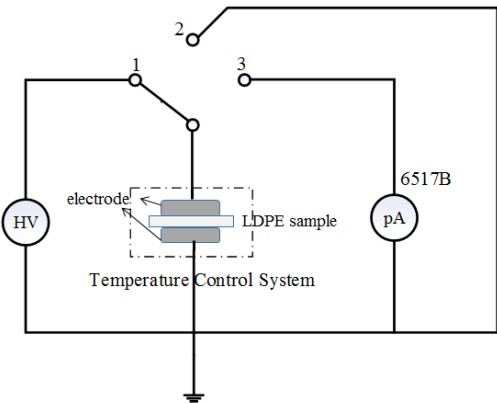

**Figure 3.** Schematic diagram of thermal stimulation current method.

## 3. Results

### 3.1. Structural Characterization of LSC

The XRD pattern of LSC is presented in Figure 4. It is observed that there are no impurity peaks from the XRD pattern, and the XRD diagram of LSC shows the characteristic of sharp peaks, indicating that the crystallization of LSC was excellent. The XRD pattern of LSC displays characteristic peaks at 2θ = (23.4°, 33.2°, 40.8°, 47.6°, 53.5°, 59.1°, 69.6°, 79.2°, 83.7°, and 88.2°,) which correspond to the planes of (012), (110), (202), (024), (122), (300), (220), (134), (042), and (404) simultaneously, consistent with the standard reference data (JCPDF:89-5719).

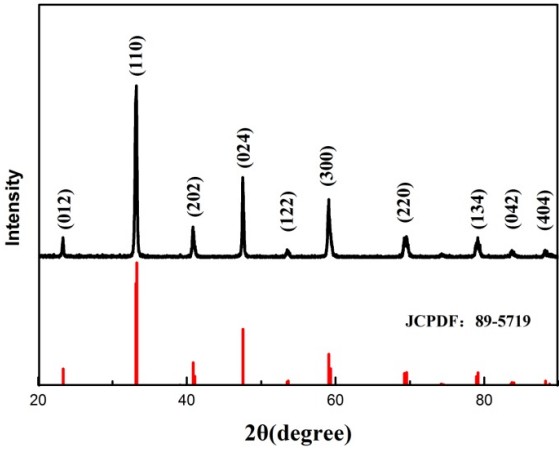

**Figure 4.** X-ray diffraction (XRD) patterns of LSC.

### 3.2. SEM of LSC

The SEM images of the LSC before ball milling and after ball milling for 10 and 20 h are shown in Figure 5. It can be seen from Figure 5 that the particle size of the nanoparticles decreases with increasing ball milling time. The LSC without ball milling is composed of particles with a particle size of about 300 nm. In Figure 5a, the grains of LSC powder are bonded together. After ball milling, the particles originally bonded together are dispersed. From Figure 5b, it can be seen that after ball milling for 10 h, the size of LSC particles is distributed around 400–700 nm. After ball milling for 20 h, the size of LSC particles is 300 nm. The particles bonded after ball milling are dispersed, and the particle size distribution is more uniform. This facilitates the preparation of a smooth semiconductive shielding layer.

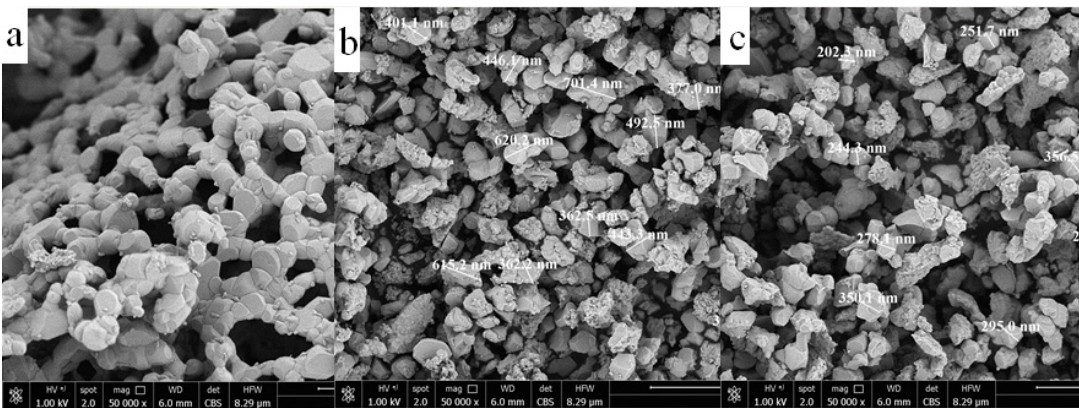

**Figure 5.** Scanning electron microscopy (SEM) images of LSC: (**a**) Ball milling (BM)-LSC (0 h); (**b**) BM-LSC (10 h); (**c**) BM-LSC (20 h).

### 3.3. SEM and TEM of the Semiconductive Shielding

The dispersion of nanoparticles in the matrix polymer can be observed by the SEM of Figure 6 and the TEM of Figure 7. Figure 6a–c shows the fracture surface SEM images of composite nanomaterials with an LSC content of 0%, 1%, and 5%, respectively. The white spots in Figure 6 are the LSC nanoparticles. It can be seen from Figure 6b that the nanoparticles are uniformly dispersed in the matrix and the white spots in Figure 6b,c increase as the LSC content increases. Figure 7 shows that, the contrast degree of the carbon black particles in the polymer matrix are light, and the black particles with deep contrast are LSC particles. It can be seen from the figure that the size of the black particles acts at several hundred nm, which matches the SEM image of the LSC particles in Figure 5. Figure 7 shows that carbon black particles fill the matrix polymer and form conductive channels. LSC particles

are uniformly dispersed in the polymer matrix. However, the nanoparticles are prone to agglomeration when the LSC concentration is high.

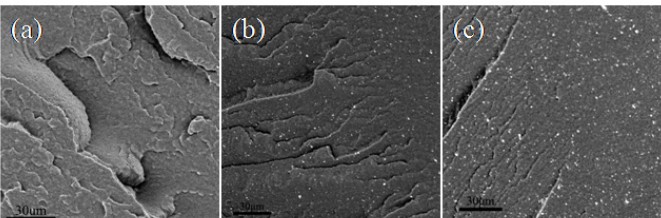

**Figure 6.** SEM of sections of non-semiconducting shielding materials with different LSC contents: (**a**) 0 wt% LSC; (**b**) 1 wt% LSC; (**c**) 5 wt% LSC.

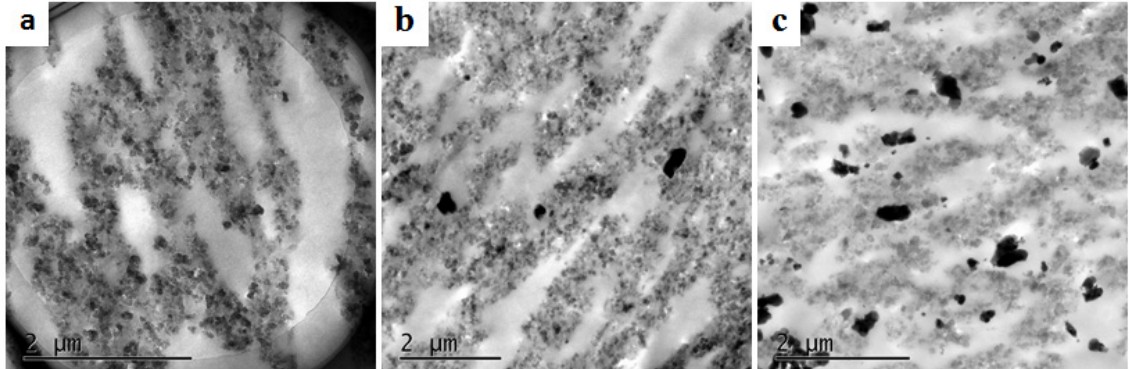

**Figure 7.** Transmission electron microscopy (TEM) micrographs of the semiconductive materials with different LSC contents: (**a**) 0 wt% LSC; (**b**) 0.5 wt% LSC; (**c**) 3 wt% LSC.

*3.4. Electrical Properties of the Semiconductive Shielding*

Figure 8 shows that curve of resistivity versus temperature for the semiconductive layer containing different mass fractions of LSC. Figure 9 shows the resistivity curve of semiconductive materials with different LSC contents at 383 K. It can be seen from Figure 9 that at 383 K, the resistivity of semiconducting shielding material without LSC doping is 798 $\rho/\Omega \cdot cm$ and when the LSC doping amount is 1 wt%, the resistivity is 128.5 $\rho/\Omega$ cm, decreased by 83.9%. Some researchers have added $SrFe_{16}O_{19}$ to semiconductor shielding materials and tested their resistivity. The resistivity of semiconductive materials with $SrFe_{16}O_{19}$ doping of 1 wt% and 5 wt% is similar to that without $SrFe_{16}O_{19}$ doping. When the doping amount of $SrFe_{16}O_{19}$ is 30 wt%, the resistivity of semiconductive materials is more than $10^3$ at 383 K [26]. We can see that the resistivity demonstrates a slow rising tendency with temperature before the temperature is below 343 K. Meanwhile, there is a huge transition in the resistivity value of the semiconductive composites without added LSC after the temperature exceeds 343 K. In other words, the semiconductive layer without added LSC possesses a significant PTC effect. Since the electrical conductivity of the LSC increases with increasing temperature, the semiconductive layer to which LSC is added still has good electrical conductivity at high temperatures. It can be seen from Figure 9 that at 383 K, the resistivity of semiconductive materials with 1 wt% LSC doping is greatly reduced compared with that without LSC. Therefore, the addition of LSC can improve the PTC effect of the semiconductive composites so that it still meets the resistivity requirements of the semiconductive layer at high temperatures. In particular, the semiconductive layer with a 1% LSC presents good electrical conductivity at high temperatures. This might be attributed to the distortion of the crystal structure of Sr-doped $LaCoO_3$, the lattice spacing becomes larger, and the amount of O vacancies increases, providing more conductive channels for carrier transport.

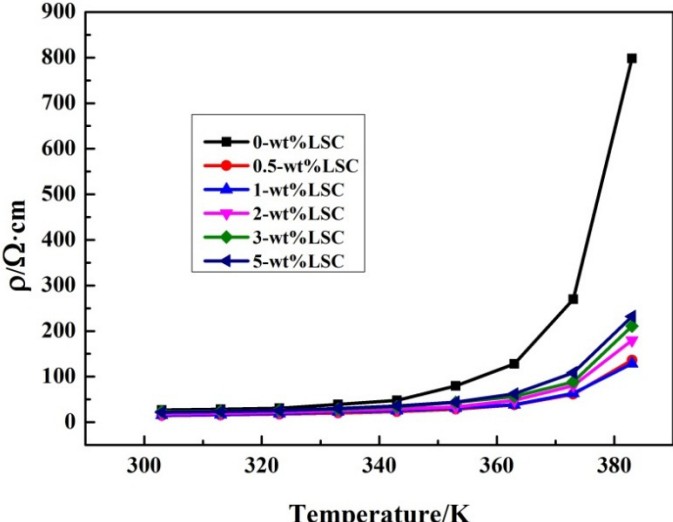

**Figure 8.** Resistivity of the semiconductive composites with different mass fractions of LSC as a function of temperature.

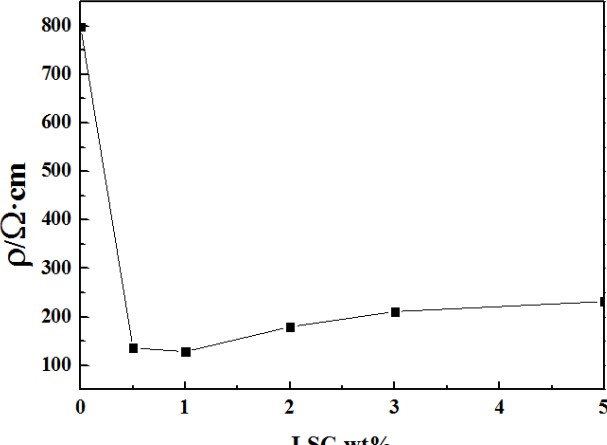

**Figure 9.** Resistivity curves of semiconductive materials with different LSC contents at 383 K.

The formula for calculating the strength of the polymer's positive temperature coefficient:

$$\alpha = \lg \frac{\rho_{v(\max)}}{\rho_{v(\min)}}$$

The calculated PTC strengths of semiconductive composites with LSC doping contents of 0%, 0.5 wt%, 1 wt%, 2 wt%, 3 wt% and 5 wt% were 1.47, 0.96, 0.91, 0.99, 0.95, 1.02. Compared with the PTC strength of the semiconductive materials without LSC, the PTC strength of the semiconductive materials with 1 wt% LSC content decreased by 38.1%, which indicated that the addition of LSC has a significant weakening effect on the PTC effect of the system, which is related to the increase in the conductivity of the ionic conductor with the increase in temperature. As the temperature increases, the number of carriers in the LSC increases, and the mobility of the carriers increases. Therefore, the PTC effect of the LSC/CB/LDPE/EVA composites is weaker. With the increase in LSC content, the PTC strength of nanocomposites decreases first and then increases. Because of the agglomeration of LSC in semiconductive materials, part of the carbon black conductive network in the composites is disconnected, thus, the PTC strength of semiconductive materials increases when LSC content is high.

### 3.5. Depolarization Current Properties

Figure 10 presents the depolarization current of the insulating layer when nanocomposites with different LSC contents were used as semiconductive layers. Figure 10a shows the depolarization current in LDPE at a 10 kV/mm DC field. It can be seen that the depolarization current increases first and then decreases with increasing temperature. The peak value of the current of all samples appeared at 330–340 K under 10 kV/mm DC field, which indicates that the trap levels are basically the same. At high loading levels, the peak value of the depolarization current of the LDPE increases as the LSC content in the semiconductive layer increases. Figure 10b,c shows the depolarization current of LDPE at 30 and 40 kV/mm. The depolarization current increases as the electric field increases, mainly because of the increased charge injection under a strong electric field. At the same time, the position of the peak moves toward the high temperature direction, mainly because the depth of charge injection increases as the electric field strength increases.

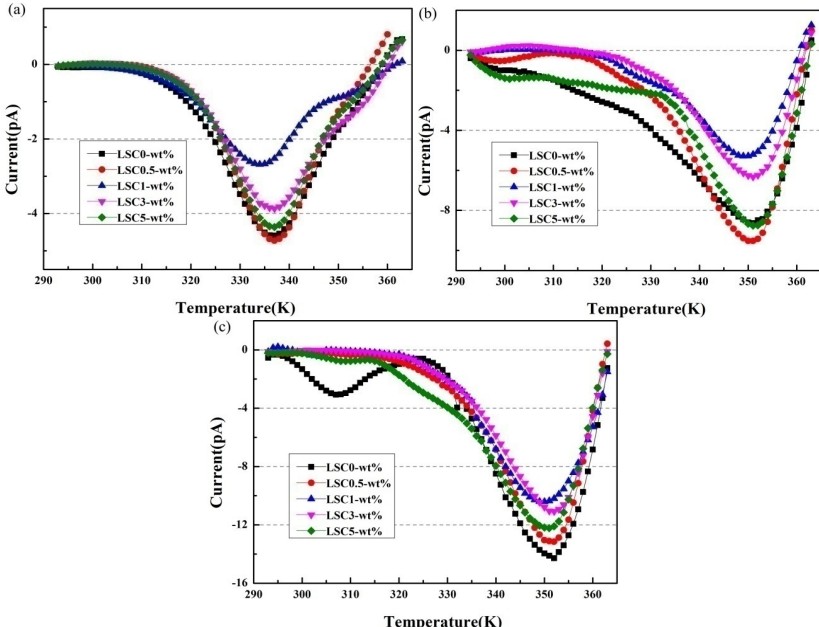

**Figure 10.** Thermal stimulation current of LDPE when different semiconductive layers are used as electrodes, the applied electric field was: (**a**) 10 kV/mm, (**b**) 30 kV/mm, (**c**) 40 kV/mm at room temperature.

The depolarization current peak that appears between 300 and 320 K in Figure 10c is due to the dipole polarization of small molecular chains and polar groups in LDPE.

The total trap charge can be calculated according to the TSDC curves. Figure 11 shows the amount of trap charge in LDPE when a composite with different LSC contents is used as a semiconductive layer. It can be concluded from Figure 11 that the effect of suppressing space charge injection when the composite material with an LSC content of 1% is used as the semiconductive layer is the most obvious. When the composites without LSC were used as the semiconductive layer, the charge amount in the insulating sample is $1.35 \times 10^{-9}$, $3.26 \times 10^{-9}$, and $4.26 \times 10^{-9}$, respectively, under 10, 30 and 40 kV/mm DC electric fields. For LSC content with 1 wt%, the charge of the insulating layer decreased to $0.75 \times 10^{-9}$, $1.34 \times 10^{-9}$, and $2.75 \times 10^{-9}$, respectively, decreasing by 44.4%, 58.9%, and 35.7%. When the LSC concentration in the semiconductive composites is high, the trap charge amount in the LDPE increases. The reason might be that the agglomeration of nanoparticles causes the surface roughness of the nanocomposite to increase, resulting in electric field distortion.

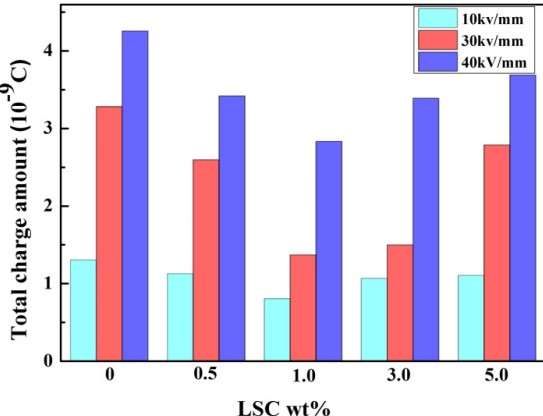

**Figure 11.** The total amount of charge in the LDPE when the composite with different LSC content acts as a semiconductive layer.

In general, when the composite material with a 1% LSC content is used as a semiconductive layer, the peak value of the depolarization current is the smallest. The depolarization currents have the same tendency at different polarization voltages.

### 3.6. Space Charge Distribution

The space charge distribution of LDPE under a 10 kV/mm and a 40 kV/mm DC electric field within 30 min at room temperature is shown in Figures 12 and 13. It can be seen from Figure 12a that the accumulation of the homocharge is observed near the cathode and the anode in the LDPE when the semiconductive layer is not added to with LSC. Among them, the heterocharge is derived from the ionization of the crosslinked byproducts and the ionization of the impurities, and the homocharge is derived from the injection of the electrodes. It can be seen from Figures 12c and 13c that there is almost no accumulation of the homo charge at the cathode. However when the content of LSC in the semiconductive layer exceeds 1%, as the LSC content increases, the space charge injection in the LDPE increases; that is, the inhibition effect of the semiconductive layer is weakened, which may be related to the agglomeration of the LSC. It can be inferred that semiconductive materials with an LSC content of 1% can suppress the injection of space charge. Due to the scattering effect at the interface between the nanoparticles and the polymer, the mean free path of electrons is increased and the migration rate of electrons is reduced.

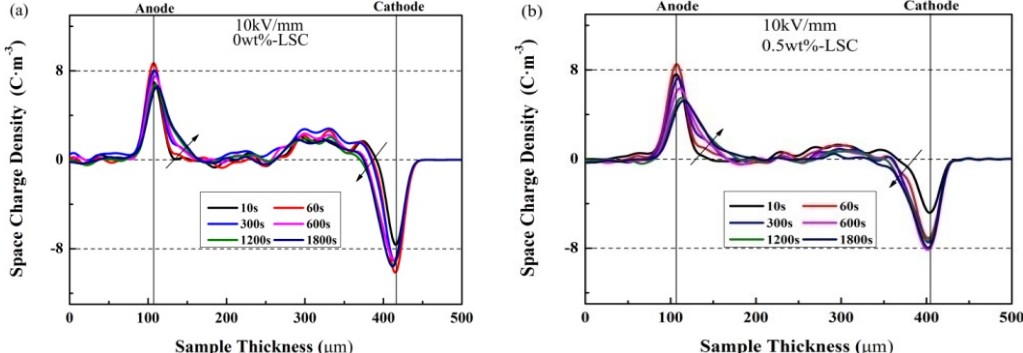

**Figure 12.** *Cont.*

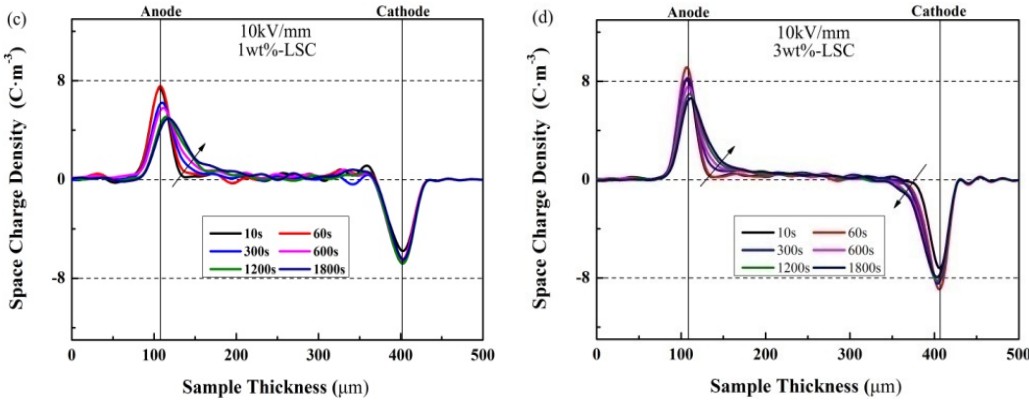

**Figure 12.** Space charge distribution of LDPE when the composite with different LSC contents acts as a semiconductive layer under 10 kV/mm DC electric field.

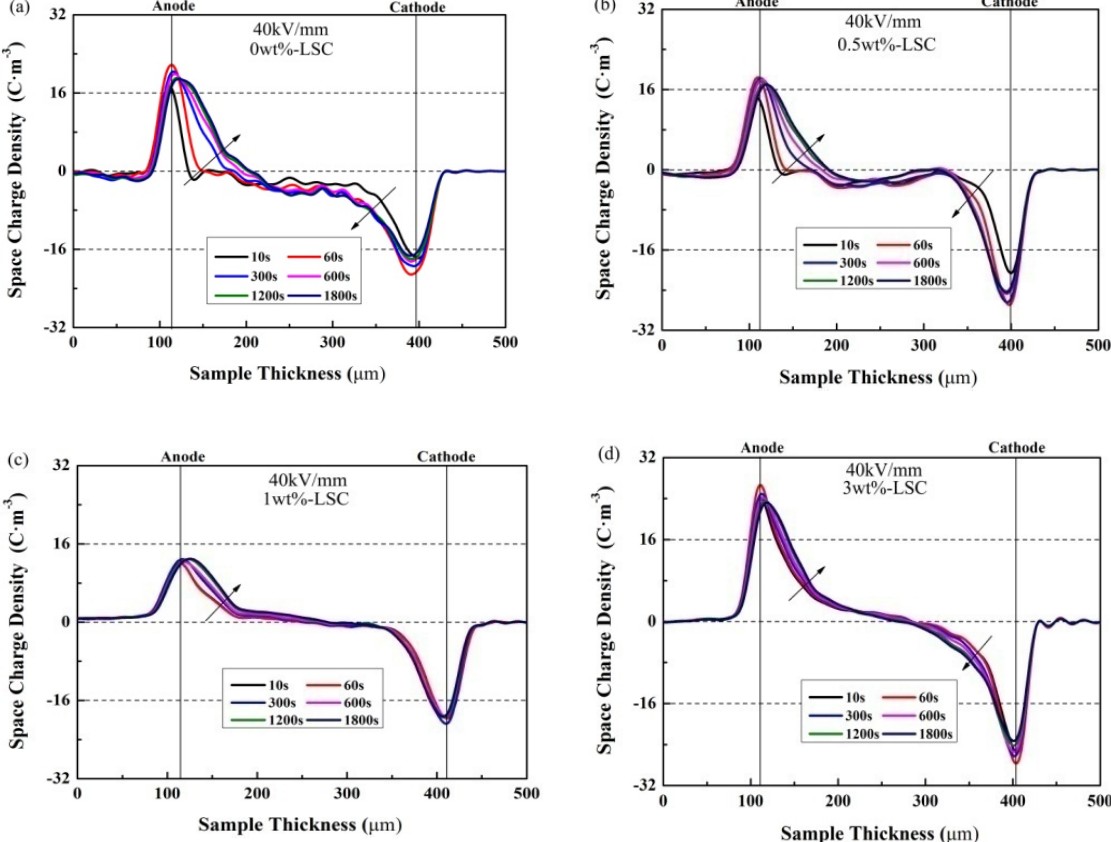

**Figure 13.** Space charge distribution of LDPE when the composite with different LSC contents acts as a semiconductive layer under 40 kV/mm DC electric field.

When the composite without LSC is used as the semiconductive layer, the maximum charge density near the cathode and anode is 11.46 and 9.37 C/m$^3$ respectively under a 10 kV/mm DC electric field. After doping by LSC with 1 wt%, the interface charge near the cathode and the anode is reduced to 6.53 and 7.76 C/m$^3$. The maximum charge density near the two electrodes is 22.15 and 21.36 C/m$^3$ under a 40 kV/mm DC electric field. When the semiconductive layer is doped with 1 wt% LSC, the interface charge reduced to 12.77 and 20.89 C/m$^3$, respectively. When the charge is injected from the metal electrode to the insulating layer, it passes through the semiconductive layer, and the charge receives the Coulomb effect of the LSC particles in the semiconducting shielding layer, so that part of the charge cannot be injected into the insulating layer through the semiconductive layer, thus reducing the charge injection in the insulating layer.

## 4. Discussion

LaCoO$_3$ has a typical perovskite structure. When Sr$^{2+}$ is added into the perovskite lattice to replace La$^{3+}$, the net electric imbalance will be caused. In order to compensate the net electric imbalance, oxygen vacancy will be generated in the lattice to bring many holes to achieve the charge balance, and oxygen vacancy is allowed to transfer through the perovskite lattice [30]. When the charge is injected from the metal electrode into the insulating layer, it needs to pass through the semiconductive layer. The oxygen vacancy of LSC crystal in the semiconductive layer has electrostatic attraction to the charge, which hinders the movement of the charge, making it difficult for the charge to be injected into the insulating layer through the semiconductive layer, thus reducing the charge injection in the insulating layer.

On the other hand, in ionic crystals, alternating charged plane stacking can generate divergent electrostatic energy, which makes the oxide surface polar. This polar surface is electrostatically unstable, and surface charge must be compensated by surface reconstruction or charged defect accumulation [31]. When the charge is injected from the metal electrode to the insulating layer, it passes through the semiconductive layer. Under the action of electric field, due to the polarity of LSC particle surface, the ions of LSC crystal will move relatively, which will cause polarization, and then lead to the interaction between the polarization field and the charge, thus reducing the charge injection in the insulator. In 1993, Landau proposed that electrons could trap themselves in the deformed lattice [32]. In 2002, Iwanaga et al. observed trapped electrons and holes in PbBr2 crystal [33]. When electrons change from free-form to self-trapped, their mobility will change obviously. As shown in Figure 14, if electrons are injected into the lattice, due to the effect of electrons on the crystal lattice, the surrounding crystal lattice is distorted, causing the positive ions around it to move closer to the electrons, and the negative ions to move far away, which is called a "polarized cloud". The polaron is a combination of electrons and a polarized cloud around it. As the electrons move to drag the surrounding polarized clouds, the mass increases and the migration rate decreases. Lattice deformation can bind electrons, thereby, the injection of electrons from the metal electrode to the insulating layer was suppressed.

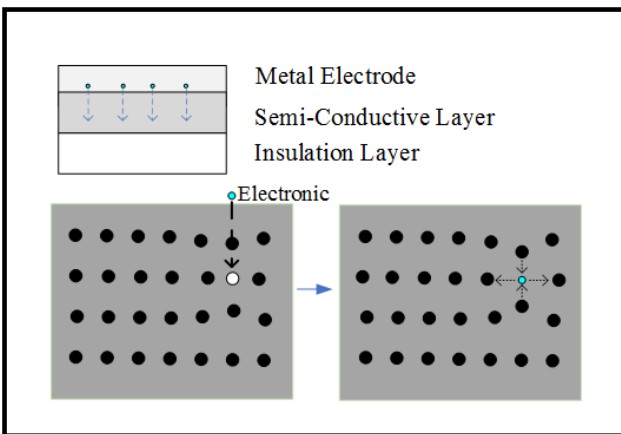

**Figure 14.** Schematic diagram of electron trapping.

## 5. Conclusions

In this paper, semiconductive layers with different contents of LSC were prepared by melt blending. The appearance and resistivity of the nanocomposites and their effects on space charge injection of insulating layers were studied. The conclusions are drawn as follows:

1.　When the LSC content in the semiconductive composites is low, the nanoparticles are uniformly dispersed in the matrix, and when the content of the nanoparticles increases, agglomeration occurs.

2. The addition of LSC can suppress the PTC effect of the semiconducting layer. When the LSC content is 1 wt%, the PTC strength of semiconducting shielding layer decreased from 1.47 to 0.99, decreasing by 38.1%. This is because the LSC doped in semiconductive materials is an ionic conductor, and the mobility of carriers increases with the increase in temperature.

3. The experimental results show that when the doping amount of LSC is 1 wt%, the charge amount in the insulating sample is the smallest, which is $0.75 \times 10^{-9}$, $1.34 \times 10^{-9}$, and $2.75 \times 10^{-9}$, respectively, decreasing by 44.4%, 58.9%, and 35.7%. This is because the charge is subjected to the Coulomb force of the LSC particles in the semiconductive layer, which reduces the charge injection from the metal electrode to the insulating layer.

**Author Contributions:** Conceptualization, C.H. and Y.C.; methodology, H.Y.; software, X.G. and S.Z.; validation, Y.W., G.L. and M.X.; writing—original draft preparation, Y.C.; writing—review and editing, C.H. and Y.C.; funding acquisition, Q.L. and Z.X. All authors read and agreed to the published version of the manuscript.

**Funding:** This research was funded by [the State Key Laboratory of Advanced Power Transmission Technology] grant number [SGGR0000DWJS1800561] and [State Grid Shandong Electric Power Research Institute] grant number [SGSDDK00KJJS1900329].

**Acknowledgments:** This research was funded by [the State Key Laboratory of Advanced Power Transmission Technology] grant number [SGGR0000DWJS1800561] and [State Grid Shandong Electric Power Research Institute] grant number [SGSDDK00KJJS1900329].

**Conflicts of Interest:** The authors declare no conflict of interest.

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
