# Peer review of "Effect of Ionic Conductors on the Suppression of PTC and Carrier Emission of Semiconductive Composites"

_applsci, doi:10.3390/app10082915_

Round 1

Reviewer 1 Report

The paper focuses on the topic of the effect of a filler on semicon performance (conductivity at increasing temperatures and its influence on charge injection).

The methodology is more than sufficient for this investigation, and results are clear.

However, the justifications that authors are claiming are often not supported by either other measurements, or other studies.

Grammar should be improved.

I would recommend its publication after mandatory revisions.

Further comments are given in the attachements.

Reviewer 2 Report

The manuscript describes the use of a perovskite, particularly La0.6Sr0.4CoO3, as a filler to polymer matrix, to lead to the formation of a composite with decreased positive-temperature-coefficient (PTC) effect. The main weakness of the manuscript is the fact that it describes only the results, with almost no discussion and no comparison with the literature data. There are many claims that are meaningless without a broad context, e.g. what is “good electrical conductivity at high temperatures”? Is any value, which is lower than the control, enough? I don’t think so. Therefore, I would strongly recommend to expand the “Discussion” section.

Moreover, there are many grammar errors, typos and inconsistencies all over the manuscript, e.g. the Authors sometimes use the dash and sometimes not (high voltage vs. high-voltage, positive temperature coefficient vs. positive-temperature-coefficient), unnecessary capital letters in lines 29-30 and 50, grammar error in line 40: “This work using CB cofilled (...)”, line 67-68 “citric acid acts as a complex to form a metal complex” – it should be “as a ligand to form a complex”, pH instead of PH (line 68), etc.

The resistivity test (section 2.2.4) should be described in more details.

Figure 2 has black text on a dark grey background (hard to read letters).

SEM images are not well described. The particle size distribution (Fig.5.) is said to be more uniform, but there is no analysis of the sizes of particles. Also, in Fig.6. the white spots are said to increase with increase in LSC content – no analysis is provided. The claim about agglomeration of particles is, therefore, not supported by data. I would not say that particles in Fig.7. are uniformly dispersed. Moreover, I believe that for the comprehensive discussion of data, TEM images should be taken for all composites (having various amounts of LSC  filler).

Therefore, I would recommend major revision, with the particular focus on expanding discussion of the results.

Round 2

Reviewer 2 Report

The "Discussion" section could still be improved. PTC strength of the described materials could be compared with literature data.
